# MATHEMATICAL REASONING VIA SELF-SUPERVISED SKIP-TREE TRAINING

**Markus N. Rabe**
Google Research
mrabe@google.com

**Dennis Lee**
Google Research
ldennis@google.com

**Kshitij Bansal**
Google Research
kbk@google.com

**Christian Szegedy**
Google Research
szegedy@google.com

## ABSTRACT

We demonstrate that self-supervised language modeling applied to mathematical formulas enables logical reasoning. To measure the logical reasoning abilities of language models, we formulate several evaluation (downstream) tasks, such as inferring types, suggesting missing assumptions, and completing equalities. For training language models for formal mathematics, we propose a novel skip-tree task. We find that models trained on the skip-tree task show surprisingly strong mathematical reasoning abilities, and outperform models trained on standard skip-sequence tasks. We also analyze the models' ability to formulate new conjectures by measuring how often the predictions are provable and useful in other proofs.

## 1 INTRODUCTION

Language modeling using Transformers (Vaswani et al., 2017) has been hugely successful for applications like translation and text generation. Models like GPT are able to generate news articles and stories given just an abstract (Radford et al., 2018). These models are usually (pre-)trained on a proxy task, such as predicting missing words in the case of BERT (Devlin et al., 2019), before fine tuning the models on more specific (downstream) tasks such as machine translation and question-answering. These proxy tasks are not reliant on labels, and thus can be trained on large corpora of unlabeled data. Recently, however, we have seen successful demonstrations of language modeling using only self-supervised training without any fine tuning (Brown et al., 2020).

In this work, we extend this line of thought and demonstrate that purely self-supervised training can even lead to mathematical reasoning abilities. This represents a major departure from prior work in deep learning for mathematics, which has focused on learning directly on logical reasoning tasks, such as predicting the proof steps or premises or assignments. These approaches require *labeled* data, which is hard to come by and typically very limited in size. In contrast, our language modeling approach to mathematics allows us to train on *unlabeled* mathematical expressions. We start with the HOList dataset (Bansal et al., 2019), which spans a wide range of mathematical topics, including topology, multivariate calculus, real and complex analysis, geometric algebra, and measure theory, formalized in the HOL Light proof assistant (Harrison, 1996).

We find that training a language model on all mathematical expressions in this dataset leads to surprisingly strong mathematical reasoning capabilities. We believe that this opens the door to different kinds of neural theorem provers, which do not only search through a well-defined search space of tactics and premises, but which are capable to generating their own lemmas and could even come up with a new Ansatz requiring a creative substitution.

For self-supervised training on mathematical expressions, we propose a novel skip-tree task, which is a specialization of the skip-sequence task that respects the tree structure of expressions. We show that models trained on the skip-tree task significantly outperform those trained on the skip-sequence task, which is the state of the art for sequence to sequence models for natural language.

Reasoning can refer to a wide range of abilities, and thus we measure the mathematical reasoning abilities of language models on a variety of tasks, including mechanical derivations, such as type inference, and also creative tasks, such as predicting under which assumptions a statement is true. As we want to study what reasoning capabilities can be acquired just through self-supervised training, we do not employ fine-tuning on these tasks. Instead, we designed the tasks to be syntactically similar to the training task, such that the language model may produce correct answers.

An advantage of formal language compared to natural language is that we can attempt to automatically evaluate statements. That is, we can let our language models produce *conjectures*, which we then try to prove using the DeepHOL theorem prover (Bansal et al., 2019; 2020). Besides evaluating the *provability* of the produced statements, we go one step further and evaluate their *usefulness*, by measuring how many times they are used as premises in proofs of other theorems.

Our contributions are as follows:

1. We show that self-supervised training on mathematical formulas alone leads to logical reasoning capabilities.

2. We introduce a new skip-tree training task that outperforms the state-of-the-art skip-sequence training. We also introduce several evaluation tasks that are subsumed by skip-tree training (i.e. predict a missing subexpression), but test specific logical reasoning abilities to make the performance of the models interpretable.

3. We suggest a way to create and evaluate mathematical conjectures using existing neural theorem provers.

The remainder of this paper is structured as follows: First, we review related work on language modeling and deep learning for mathematics in Section 2. Then, in Section 3 we discuss the source corpus of formal mathematical statements from which we generate our training data. In Section 4, we present the skip-tree training task, as well as several variations that we used in our ablation studies. We present the evaluation tasks in Section 5, discuss our experimental findings in Section 6, and conclude in Section 7.

## 2 RELATED WORK

Recently, we have seen a series of rapid improvements in language modeling stemming from better pretraining tasks (Devlin et al., 2019; Zhang et al., 2019; Song et al., 2019; Dong et al., 2019; Raffel et al., 2019; Conneau and Lample, 2019). BERT (Devlin et al., 2019) is a pretraining task for Transformers (Vaswani et al., 2017), which masks out a certain fraction of the input tokens that the model then has to predict. UniLM uses multiple pretraining tasks (Dong et al., 2019). One of them is a sequence-to-sequence task; to predict the next sentence from the previous sentence. MASS and SpanBERT consider a generalized sequence-to-sequence pretraining task, which is to predict a masked out subsequence of the input (Song et al., 2019; Joshi et al., 2020). However, both MASS and SpanBERT reveal the length of the sequence to predict as they replace it by a number of mask tokens equal to the length of the sequence.

T5 introduced a generalization of sequence-to-sequence pretraining tasks that is crucial to our work (Raffel et al., 2019). They replace the subsequence (or multiple subsequences) to be predicted by a single token (not a number of mask tokens equal to the length of the subsequence, as in MASS). Zhang et al. (2019) additionally exploit the sentence structure of natural language. They suggest the pretraining task Pegasus, which masks out entire sentences of a given text, and additionally masks out randomly selected tokens in the remaining text (or alternatively replace them by other tokens). In a similar way Pegasus' exploitation of the sentence structure of natural language, our skip-tree task exploits the tree structure of formal expressions. Zhang et al. (2019) also suggest sampling the sentences to be masked with the help of ROUGE1-F1 (Lin, 2004).

We work with the HOList dataset by Bansal et al. (2019), which is closely related to the Flyspeck dataset by Kaliszyk and Urban (2014). There are other datasets which might be suitable for our approach as well, including proofs extracted from HOL4 (Gauthier et al., 2017), and from Coq (Huang et al., 2019; Yang and Deng, 2019; Sanchez-Stern et al., 2019).

Most previous works that apply sequence-to-sequence models to logics have focused on specific logical tasks in *supervised* training settings (e.g. Piotrowski and Urban (2020a)). In contrast, we train language models on a self-supervised proxy task that does not require labeled data and can thus be applied to almost any source of mathematical expressions. Lample and Charton (2020) use a Transformer model for symbolic integration. They train their model directly on the the task to produce the integral of a given expression. To generate training data, their approach needs a classical algorithm to compute the derivative of the expressions. Finkbeiner et al. (2020) explore the generalization properties of the Transformer architecture predicting the solutions to SAT formulas and temporal logic, but require a data generator that can solve formulas, which is currently not feasible for higher-order logic. Piotrowski et al. (2019) train RNNs on individual logical reasoning steps, such as substitutions, using a dataset of rewrites on polynomials extracted from Prover9. Wang et al. (2018) translate between synthetic descriptions in natural language and formal mathematics on a dataset generated with Mizar.

Self-supervised training techniques for formal mathematics have received much less attention. Wang et al. (2020) apply recent self-supervised translation techniques by Lample et al. (2018) to align formal and informal statements. Very recently, Li et al. (2020) and Polu and Sutskever (2020) applied language modeling to proofs of formal mathematics. In contrast, this work focuses on measuring reasoning abilities on mathematical statements (not necessarily proofs) achieved through self-supervised training only.

Independently from our work, Urban and Jakubův (2020) presented initial experiments on applying self-supervised language modeling to formal mathematics in order to produce conjectures. However, they only evaluate the learned models through the truth of the produced conjectures, while we also consider several reasoning tasks and measure the usefulness of conjectures. Earlier methods to produce conjectures were limited in scope. For example, Piotrowski and Urban (2020b) propose a method to predict the next literal in an automated theorem prover using recurrent neural networks after supervised training. Prior to that Gauthier et al. (2016) relied only on statistical approaches to produce conjectures. Applying natural language techniques to formal mathematics has a long history. Already in 2004, Cairns (2004) applied information retrieval based on latent semantics to improve over search for keywords, and Urban (2004) formulated the intention to learn from large amounts of data in formalized mathematics.

Transformer models for program understanding have focused on providing inductive biases in the architecture (Shiv and Quirk, 2019; Hellendoorn et al., 2020), whereas this work suggests to use a modified language modeling proxy task.

## 3 DATASET

We start from the HOList dataset introduced by Bansal et al. (2019). The complete dataset includes 29465 theorems and their proofs. We here consider only the "core" and "complex" datasets which comprise 18943 theorems, 637 definitions and 603,950 proof steps. The theorems and proofs were written (by humans) using the HOL Light proof assistant, and span various areas of mathematics such as set theory, arithmetic, linear algebra, topology, and multivariate complex analysis. Each proof starts with the theorem statement as a *proof goal*. For each goal, the dataset contains a *tactic* that a human applied to it, which then produces a list of subgoals. Most tactics have arguments, such as previously proven theorems (also called *premises*), which are invoked by that tactic. From this dataset we extract all theorem statements as well as all intermediate proof goals.

We use S-expressions to represent statements. For example, consider $x = y$, which in s-expression syntax is represented as follows:

```
(a (a (c (fun (A) (fun (A) (bool))) =) (v A x)) (v A y))
```

Each subexpression here is either a leaf or a triple. The first element of these triples indicates their kind: `a` indicates function applications, `c` indicates constants (i.e. symbols that have been defined in the formal system), `v` indicates a variable, and finally `fun` indicates a function type. The equality operator "=" is represented by `(c (fun (A) (fun (A) (bool))) =)`, which indicates that it is a constant that has type `(fun (A) (fun (A) (bool)))`. This type indicate it is a curried function with two arguments of arbitrary type, indicated by the generic type variable `A`, and returns a bool. The variables $x$ and $y$ are represented as `(v A x)` and `(v A y)`. The `v` indicates that this

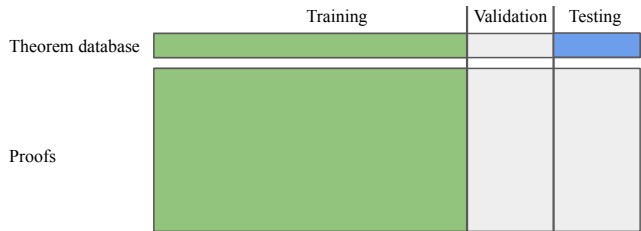

Figure 1: We use the theorems and proofs of the training split, marked in green, for training. For measuring the final performance of our evaluation tasks, we only used the theorems of the test set, marked in blue. This ensures that the models have never seen the statements from which the evaluation tasks are derived.

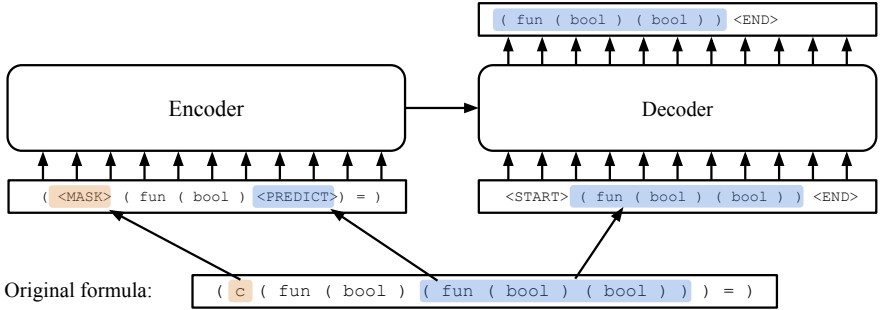

Figure 2: The skip-tree training task for the example of the equality operator on boolean constants (original formula). In this example we assume that a part of the type was sampled to be the subexpression to be predicted, and that subexpression `c` was sampled to be masked out additionally. Note the input to the decoder is shifted to the right, such that the next token prediction task yields the target sequence.

subexpression is a variable, and $A$ is a type variable, which indicates that $x$ and $y$ could have any type. Since both $x$ and $y$ use the same type variable $A$, they must have the same type.

In our dataset, we wrap each S-expression derived from a theorem in a `<theorem>` node, and all other expression in a `<goal>` node. That is, an expression `(c (fun ...)   ...)` is represented now as `(<goal> (c (fun ...)   ...))`.

We use the same split as used in HOList (Bansal et al., 2019). They first split the theorems into training, validation, and test, and then assign all statements in entire proof of each theorem to the same split as the theorem. This means that we have used the proof of 11,655 theorems in the training split of the core and complex libraries. This avoids partially revealing the proofs of theorems in the validation and test sets during training. We derive all training data from the theorems and proofs in the training set, and use only the theorems (not the proofs) for the evaluation tasks. This addresses the possibility that some proof steps for training theorems and for validation theorems might be shared. In Figure 1 we depict our choice of training and evaluation data.

## 4 SKIP-TREE TRAINING

In this section we define the skip-tree training task. We parse a given mathematical statement into a tree of subexpressions, and replace one of the subexpressions by a <PREDICT> token. The task is to predict the subexpression replaced by <PREDICT>. See Figure 2 for an example.

For training, the trees are converted back to a sequence of tokens; the target sequence is extended by a <START> token in the front and an <END> token in the back. We filter out training examples that

- are derived from an input sequence that is longer than 10k characters as our models would see only a small prefix of the input sequence (encoder length is 1024 tokens), and

- exceed the length of the decoder (512 tokens).

While the second criterion drops only a negligible 0.1% of the training examples, the first criterion drops around 32% of the training examples. We confirmed that not dropping those training examples does not significantly change the results (see the ablation study titled "unfiltered").

The evaluation sets were also filtered, but only 0.16% of the evaluation examples were dropped.

**Additional masked subexpressions.** In addition to the subexpression to be masked out by <PREDICT>, we select $k = 2$ subexpressions to be masked out by a different mask token <MASK>. In contrast to the <PREDICT> token, we replace *all occurrences* of these subexpressions by the <MASK> token. Note that it can happen that the subexpressions we want to replace by the <MASK> tokens overlap with each other or with the subexpression replaced by the <PREDICT> token. In this case, we give the highest preference to the <PREDICT> token, and then in decreasing order of size for the expression to be replaced by the <MASK> tokens.

The subexpressions masked by <MASK> do not have to be predicted, they are only hidden. They make the model tolerant to having only partial information. As a side-effect masking out additional terms makes the tasks harder and shorter.

**Distributions of subexpressions.** Sampling subexpressions uniformly at random results in very short sequences to be predicted: since our trees are mostly ternary, two thirds of the subexpressions are leaves. Besides picking subexpressions uniformly at random, we thus experiment with weighting the subexpressions by the number of tokens they contain. We refer to these variants as "uniform" and "weighted". This results in a much more diverse set of expressions to be sampled. The experiments show that this helps on some, but not all, tasks.

**Multiple samples per statement.** We generate up to $n = 100$ training examples from each statement in the source data set by sampling different subexpressions to predict (and different <MASK> tokens). To avoid duplicates, we sample the subexpressions to predict *without replacement*. For small formulas with less than 100 subexpressions, this reduced the number of training examples we generate from them.

Our initial data, the core and complex corpus of HOList, consists of 604K mathematical statements, from which 376K statements are in the training split. After sampling and filtering out training examples that do not fit the requirements listed in the beginning of this section, we are left with around 25.8M training examples. Table 1 lists the statistics of the datasets and various ablations.

## 4.1 ABLATIONS

To verify the design choices of the skip-tree training task we generated multiple variants of the training task and trained a model on each of them.

**No mask tokens.** To answer the question of whether it helps to mask out subexpressions besides the one to predict, we generated a dataset with $k = 0$, called "skip-tree (no <MASK>)".

**Fewer samples per statement.** Instead of sampling many training examples from each formula, we could train on a fewer training examples for more epochs. We generated a smaller version with $n = 20$ of the skip-tree training data, which we call "skip-tree (small)".

**Skip-sequence.** For natural language the state-of-the-art self-supervised training task for sequence-to-sequence models is the skip-sequence task (see MASS (Song et al., 2019), SpanBERT (Joshi et al., 2020), and T5 (Raffel et al., 2019)). The skip-tree task is similar to the skip-sequence task. But instead of predicting arbitrary subsequences in the skip-sequence task, the skip-tree task makes sure that the subsequence to predict is a subexpression. For example, for the statement $a + b = c + d$, the skip-sequence task may select the subsequence "$= c+$", but the skip-tree task would only allow us to pick valid subexpressions, such as "$a + b$". Our experiments will show that this subtle difference has a dramatic impact on the ability of our language models to predict correct answers to reasoning tasks. We generated three datasets for the skip-sequence task, where we sample subsequences of different lengths (short/medium/long), with lengths up to 50, 100, and 512 tokens.

| Dataset | # examples | # tokens (input/output) | avg length (input/output) |
|---------|-----------|------------------------|---------------------------|
| Skip-tree (weighted) | 25.8M | 17.4B/1.6B | 675/61 |
| Skip-tree (uniform) | 25.7M | 18.8B/316M | 732/12 |
| Skip-tree (small) | 5.2M | 3.5B/521M | 673/100 |
| Skip-tree (unfiltered) | 37.4M | 30.6B/3.1B | 817/83 |
| Skip-tree (no <MASK>) | 25.8M | 19.4B/1.6B | 750/61 |
| Skip-sequence (long) | 19.2M | 11.9B/2.8B | 620/146 |
| Skip-sequence (medium) | 26.0M | 19.4B/884M | 744/34 |
| Skip-sequence (short) | 26.0M | 19.6B/479M | 752/18 |

Table 1: Basic statistics of the *training splits* of the data sets. Number of tokens in the training set measured before padding.

**Unfiltered.** While generating the training data, we filtered out mathematical statements that exceed 10k characters. This removed some 32% of the statements from the HOList corpus. In this ablation study we removed this filter and train on the full corpus.

## 5 EVALUATION TASKS

In this section we suggest several logical reasoning tasks on which language models can be evaluated. These tasks require different levels of logical reasoning, ranging from mostly mechanical tasks, such as type inference, to more creative tasks, such as predicting missing assumptions. We intentionally define them to have the same format as the training task, i.e. predict a missing part of a larger expression, as this allows us to test models without fine-tuning.

However, the evaluation tasks are out-of-distribution in two ways: First, we generate them from the theorems (excluding proofs) from the validation/test sets. This ensures that the model has never seen the theorems from which we generated these evaluation tasks, nor has it seen the proofs of these theorems. This makes the tasks more challenging, and forces the models to go beyond memorization. Second, we mask out very specific elements, such as types and assumptions. This makes the results on the evaluation tasks easier to interpret. To give the interested reader a better impression of the evaluation tasks, we provide a list of randomly selected examples in Appendix E.

**Type Inference.** We generate type inference problems by omitting the typing annotation of variables, constants, and (lambda-)abstractions. We generated two variants: In the task we call "Type Inference," we replace only the selected type by the <PREDICT> token and do not mask out anything else. In the second variant we name "Hard Type Inference," we additionally replace *all other types* by the <MASK> token. The two tasks loosely correspond to the deriving the first and the last type during type inference.

Consider the following example of a "Type Inference" evaluation task:

```
(a (a (c <PREDICT> =) (v A x)) (v A y))
```

The type of the equality operator is uniquely defined, given the types of the two subterms of the equation. In this example the type could have been computed by a classical type inference algorithm.

For the "Hard Type Inference" evaluation task, the input could look as follows:

```
(a (a (c <PREDICT> =) (v <MASK> x)) (v <MASK> y))
```

Now, the type inference task is highly ambiguous. In fact, variable x could have *any* type, and the equality operator would have to adapt to the type of its arguments accordingly.

**Assumptions.** This evaluation task is to predict missing assumptions for theorems in the validation set. We extract these tasks by searching for "top-level implications" and replacing their left operand by the <PREDICT> token. We define an implication operator "$\Rightarrow$" in an expression to be a *top-level implication* if it is either the top-most operator of the expression, or occurs only under quantifiers, conjunctions, disjunctions, or on the right side of other top-level implications. This definition helps us to avoid picking assumptions in negated parts of formulas.

Note that we can have multiple top-level implications per validation theorem. Consider the abstracted example $(a \Rightarrow b) \wedge (c \Rightarrow (d \Rightarrow e))$. In this case, $a$, $c$, and $d$ are all considered to be assumptions of top-level implications.

An example from the theorem database is $x = y \Rightarrow a + x = a + y$, for which the task is to predict $x = y$ given $\texttt{<PREDICT>} \Rightarrow a + x = a + y$. (We omit the presentation of this example as an s-expression for the sake of readability.) At first, the expression to predict in this case may seem unique, but there are actually many ways to complete the task into a true statement; e.g. $y = x$ or $x = 0 \wedge y = 0$. Still, most humans would likely guess $x = y$ as it is simple and general, and because $x$ occurs before $y$ in the alphabet. To make a correct prediction, our language models thus have to not only reason which which statements are likely correct answer, but also needs to find a reasonably general statement and also know about naming conventions.

Below we give some examples of this reasoning task that we selected for their simplicity. (For a representative selection, see Appendix E.) While it is often easy to "see" that a given solution to such a task is correct, it can be non-trivial to come up with a solution in the first place. We encourage the reader to make their own predictions before looking up the ground truth in Appendix C:

- $\texttt{<PREDICT>} \Rightarrow (g \setminus \{s\}) = g$
- $\texttt{<PREDICT>} \Rightarrow (x1/y1 = x2/y2 \Leftrightarrow x1 * y2 = x2 * y1)$
- $\texttt{<PREDICT>} \Rightarrow (x \Leftrightarrow (b \vee x1) \wedge (b \vee x0))$

**Equalities.**    Similar to the task of predicting missing assumptions, we ask to predict one side of a top-level equality in this task. Again, we define top-level equalities to be any equality that occurs as the top-level operator of the formula or occurs inside quantifiers, conjunctions, disjunctions, or on the right side of implications. For example, from the theorem $\forall x. x = (x = \texttt{True})$ we extract two evaluation examples: $\forall x. \texttt{<PREDICT>} = (x = \texttt{True})$ and $\forall x. x = \texttt{<PREDICT>}$.

Again, we present some simple example tasks (in human-readable notation) and provide the ground truth as well as the model predictions in Appendix C:

- $\forall x, n \in \mathbb{N} : (x^n = 1) = \texttt{<PREDICT>}$
- $\forall m, n : n \leq m \Rightarrow m - n + n = \texttt{<PREDICT>}$
- $\forall l, m : \texttt{<PREDICT>} = \texttt{APPEND}(\texttt{REVERSE}(m), \texttt{REVERSE}(l))$

## 6    Results and Discussion

In language modeling for natural language one of the key metrics is how often the next token in the ground truth is correctly predicted. This is not an ideal measurement for formal mathematics as even a single incorrect token can invalidate the entire statement. Also, the S-expression representation is relatively lengthy and barely human-readable, so a token-level measurement does not allow us to compare our models to the natural language models in any case. *In the first part of our evaluation we therefore focus on exact (i.e. syntactic) matches of the entire predicted statement.*

We trained a Transformer architecture on the skip-tree dataset and each of the ablations for up to 1M steps (=parameter updates) with a batch size of 256. This means our models are trained for about 10 epochs (depending on dataset size). Our models have 39M trainable parameters; the hyperparameters are specified in the appendix. We trained them on an 8x8 TPU configuration, which equates to 128 cores. The training runs took between 5 and 15 hours, depending on the average length of the output sequences, which translates to up to 1.4 and 4.2 PetaFLOPs days per training run.

We measured each evaluation task on 1000 samples. The evaluation was performed on CPUs in a cloud computing framework on recent CPU architectures. Depending on the average length of the output sequences this required a different amounts of resources: (regular and hard) type inference took 4 CPU hours, equality completion took 24 CPU hours, missing assumption took 43 CPU hours. The average prediction time per token on CPU is around 60 milliseconds. Each evaluation task was repeated every 50k training steps on 1000 freshly sampled examples of the validation set. We then picked the best checkpoint (based on the results on the validation data) and evaluated it on the test set. In Table 2 we present the results of this experiment.

**Skip-tree vs skip-sequence.**    The skip-tree task and its ablations clearly dominate the skip-sequence task. One major difference between skip-tree and skip-sequence tasks is the lack of <MASK> tokens in the skip-sequence task. We therefore have to compare its performance to the "Skip-tree (no

| Dataset | Type Inference | Hard Type Inference | Assumptions | Equalities |
|---|---|---|---|---|
| Skip-tree (weighted) | **96.7%** | **94.0%** | **43.9%** | **47.4%** |
| Skip-tree (uniform) | **96.3%** | **94.2%** | 31.3% | 37.7% |
| Skip-tree (small) | **96.2%** | 90.6% | 39.3% | 36.6% |
| Skip-tree (unfiltered) | 95.8% | **94.3%** | 42.9% | **47.9%** |
| Skip-tree (no <MASK>) | 95.8% | 20.8% | 42.5% | 45.9% |
| Skip-sequence (long) | 12.6% | 0.0% | 1.5% | 1.0% |
| Skip-sequence (medium) | 66.7% | 6.6% | 8.0% | 9.9% |
| Skip-sequence (short) | 87.3% | 12.6% | 3.1% | 9.0% |

Table 2: Success rate of predicting the ground truth in a beam search of width 8 after training a model on various datasets. Bold numbers indicate results that are within 0.5% of the best result. Grayed out values indicate experiments where the training data did not include the <MASK> token but the evaluation did.

<MASK>)" ablation study to get a fair picture. The length of the skipped sequences appears to play a substantial role, with the Skip-sequence (medium), masking out sequences of up to length 100, performing best. A manual inspection of the predictions of the skip-sequence models showed they rarely parse or typecheck. It seems that the skip-sequence models consistently add surplus tokens at the end, or stop expressions too early; *they appear to be unable to correctly identify the end of the expression*.

**Impact of <MASK> tokens.** Hard Type Inference is the only evaluation task that contains the <MASK> token. Models trained on datasets without the <MASK> token perform poorly here (see grayed-out numbers in Table 2). The presence or absence of <MASK> tokens has only a minor impact on the other, as we can observe through the comparison of "Skip-tree (weighted)" and "Skip-tree (no <MASK>)".

**Multiple samples per statement.** For each source statement in the HOList corpus we sampled $n = 100$ subexpressions to be masked out. Lowering $n$ to 20 significantly decreased the performance, as we can see in the comparison between "Skip-tree (small)" and "Skip-tree (weighted)". Sampling many subexpressions per source statement appears to be a good way to increase the number of training examples from limited source data.

**Uniform vs weighted sampling.** Sampling subexpressions weighted by their size in the skip-tree task significantly improves the performance on the harder tasks, missing assumptions and equality completion. On the Type Inference tasks, the performance is very similar. We conjecture this is because of the average size of the terms to predict is smaller for the uniform sampling strategy, which is more similar to the average size of the types to predict.

## 6.1 CONJECTURING

In the experiments above, we measured how often the models predicted the ground truth in the evaluation tasks. We now change our point of view, and examine whether the models can be used to generate new and useful conjectures. In the following, we analyze all statements produced in the Assumptions and Equalities tasks in Table 2, and we introduce a new task, which we call *free-form conjecturing*.

For free-form conjecturing we simply ask the model to produce theorems by presenting the input sequence (the "prompt"): (`<theorem> <PREDICT>`). The subexpression the models has to fill in is thus an entire theorem. We use a beam search with beam width 1024 to produce enough outputs for a meaningful evaluation.

**How often are predictions true and new?** For this measurement, we attempt to prove the conjectured statements with the DeepHOL theorem prover (Bansal et al., 2019). This gives us a *lower bound to the number of true statements*, as the version of the DeepHOL theorem prover used here can prove around 58% of the validation theorems. So we expect the estimates here to be considerably below the number of actually true statements. In Table 3 we report two numbers for each conjecturing task: The first number is the percentage of generated statements known to be provable, including exact matches, statements from the training set, and statements provable with DeepHOL. The second number is the

| Dataset | Assumptions | Equalities | Free-form Conjecturing |
|---|---|---|---|
| Skip-tree (uniform) | 30.30%/27.83% | 16.73%/13.72% | 60.16%/30.26% |
| Skip-tree (weighted) | 29.49%/25.60% | 17.38%/13.64% | 44.73%/25.18% |

Table 3: Percentage of "provable statements"/"provable **new** statements". The type inference tasks are not included as we are only interested in the predictions that do not match the ground truth. For the type inference tasks, these statements are either semantically equivalent to existing statements or statements that do not type check.

percentage of generated statements that are provable and *new* - excluding exact (syntactic) matches with the ground truth and statements from the training set.

We believe that these measurements show a significant bias towards true statements. While in some tasks, less than half of the statements were provable, there are many more ways to write a false statement than a true statement.

**Are the conjectures useful?** For some evaluation tasks, the models could "cheat" on the truth metric by making the statements *trivially* true. For example, the models can predict `False` as an assumption, or complete the missing part of an equation by making it an identity (e.g. complete `x = <PREDICT>` by predicting `x`). In fact, manual inspection revealed several such cases.

To make this measurable, we added the provable statements to the theorem database, and ran the reinforcement learning experiments of the DeepHOL theorem prover (Bansal et al., 2019) and then measured how many of the statements were used as premises. In this experiment we also make sure that the new theorems cannot be used in the proofs of their premises. In a "pruning" step DeepHOL minimizes proofs by removing each individual premise in a proof and checking if the proof still holds. This filters out statements that have no effect on the proof. Only the premises that survive this step are classified as *useful*.

We ran three reinforcement learning experiments, one for each set of conjectures produced by one the evaluation tasks. We then measured how many of the theorems generated by each task are used as a premise in one of the over 200,000 proofs found for each of the experiments. For the assumptions task, 505 of the 813 theorems were used at least once. For the equalities task and the free-form conjectures it was 831 out of 1811 and 54 out of 172, respectively. We provide usage histograms in Appendix B. While some of the most frequently used conjectures turned out to be alpha-equivalent to existing theorems in the database, we found some interesting examples among the most used conjectures produced:

- $b = a + c \Rightarrow a = b - c$.
- $\text{COUNTABLE}(\{s(n) \mid n \in \mathbb{N}\})$.
- $\forall f, s : (\forall x : x \in s \implies f(x) = \text{vec}(0)) \implies f \text{ integrable\_on } s$.

In fact, humans have used the first conjectured theorem over vector arithmetic in many proofs. However, this theorem has always been defined as a *local* lemma and thus did not make it into the theorem database. For theorems two and three in the list above, thorough manual search has revealed no closely related statement in the theorem database.

This suggest that self-supervised language models show *some* ability to produce new, useful conjectures, even without fine tuning or specialized training.

## 7 CONCLUSION

In this work, we applied the paradigms of self-supervised language modeling to formal mathematics and show that, surprisingly, this leads to mathematical reasoning capabilities. For training, we introduced a novel self-supervised skip-tree task for formal mathematics that outperforms existing training tasks used for natural language. We also suggested several evaluation tasks for measuring mathematical reasoning capabilities of language models for formal mathematics without the need of fine tuning. Finally, we explored the ability of language models to produce new conjectures by measuring how many of the new predictions are provable and useful for proving other theorems.

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

## A    HYPERPARAMETERS

We trained the Transformer models for the skip-tree tasks with these hyperparameters:

- vocabulary size: 1200
- embedding size: 128
- attention dropout: 0.1
- nonlinearity: gelu
- hidden layer dropout: 0.1
- hidden layer size: 512
- initializer range: 0.02
- intermediate size: 2048
- number of attention heads: 8
- number of hidden layers in encoder: 8
- number of hidden layers in decoder: 4
- learning rate: 1e-4

We settled with these hyperparameters after trying various alternatives. We explored encoders and decoders with up to 12 layers and various learning rates and intermediate sizes.

## B    USAGE STATISTICS OF CONJECTURES

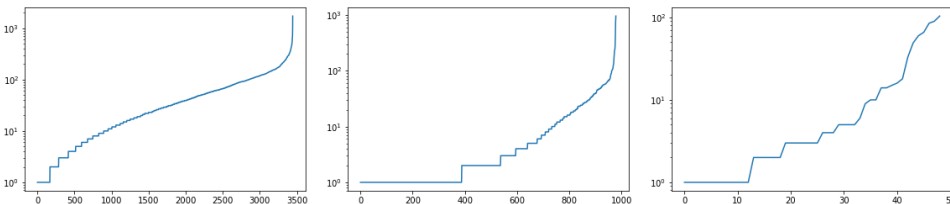

Figure 3: Histograms of premise usage of the conjectures generated through the assumptions task (left), the equality task (middle), and through free-form conjecturing (right). X-axes are the new theorems, sorted by number of usages. Y-axes indicate the number of usages on a log scale.

## C    A CLOSE LOOK AT SIMPLE EXAMPLE TASKS

**Assumptions.** In Section 5 we presented the following three examples of the task to predict missing assumptions. For the sake of readability we here discuss only the pretty printed versions. For examples in s-expression syntax, please visit Appendix E.

- <PREDICT> $\Rightarrow (g \setminus \{s\}) = g$
- <PREDICT> $\Rightarrow (x1/y1 = x2/y2 \Leftrightarrow x1 * y2 = x2 * y1)$
- <PREDICT> $\Rightarrow (x \Leftrightarrow (b \vee x1) \wedge (b \vee x0))$

The ground truth answers are as follows:

- $\neg(s \in g)$
- $0 < y1 \wedge 0 < y2$, note that $0 \neq y1 \wedge 0 \neq y2$ would be a more general assumption.
- $((b \Leftrightarrow \texttt{False}) \Rightarrow (x \Leftrightarrow x0)) \wedge (b \Leftrightarrow \texttt{True}) \Rightarrow (x \Leftrightarrow x1)$

We prompted one of our skip-tree models with these tasks. For the second and the third task, the model "skip-tree (weighted)" makes a correct prediction in the top 3 candidates in a beam search of width 8. For the first task, the model mostly produces incorrectly typed expressions: it appears to think that $s$ is a set of the same type as $g$.

**Equalities.** We presented these examples for the equality evaluation task:

- $\forall x, n \in \mathbb{N} : (x^n = 1) = \texttt{<PREDICT>}$
- $\forall m, n : n \leq m \Rightarrow m - n + n = \texttt{<PREDICT>}$
- $\forall l, m : \texttt{<PREDICT>} = \text{APPEND}(\text{REVERSE}(m), \text{REVERSE}(l))$

The ground truth for the tasks is:

- $x = 1 \vee n = 0$
- $m$
- $\text{REVERSE}(\text{APPEND}(l, m))$

Examples two and three are predicted correctly in a beam search with beam width 8. For the first example, the model almost gets it correct in two of the 8 attempts: $x = 1 \vee n = 1$, and $x = 0 \vee n = 1$. We find it surprising that the model apparently understands that there are two cases to consider, but that the exact combination of constants (1 and 0) is a challenge.

## D   MODEL PERFORMANCE BY TRAINING STEP

In Figure 4 we can see the performance of the model throughout training. We can see that the performance on validation and test is very similar, but that there is some variance. We can also observe that even after 1M steps, the model has apparently not quite converged.

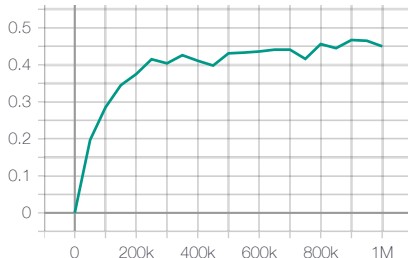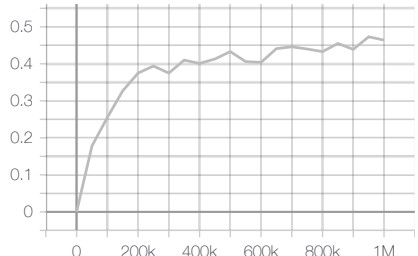

Figure 4: Performance on the Missing Assumptions task of a model trained on the Skip-tree (weighted) task. Y-axis is the training steps and X-axis is the model performance as the ratio of correctly predicted examples. Left: validation data. Right: test data.

## E   RANDOMLY SELECTED EXAMPLE TASKS

In the following, we provide a list of 5 examples for each of the evaluation tasks, sampled uniformly at random.

**Type Inference.**

```
• (<theorem> (a (c <PREDICT> !)   (l (v (fun (cart (real) ?1)
  (bool)) t) (a (c (fun (fun (fun (cart (real) ?1) (bool))
  (bool)) (bool)) !)   (l (v (fun (cart (real) ?1) (bool)) u)
  (a (a (c (fun (bool) (fun (bool) (bool))) ==>) (a (a (c
  (fun (bool) (fun (bool) (bool))) ∧) (a (c (fun (fun (cart
  (real) ?1) (bool)) (bool)) !)   (l (v (cart (real) ?1) b) (a
  (a (c (fun (bool) (fun (bool) (bool))) ∨) (a (c (fun (fun
  (cart (real) ?1) (bool)) (bool)) ?)   (l (v (cart (real) ?1)
  w) (a (a (c (fun (bool) (fun (bool) (bool))) ∧) (a (a (c
  (fun (cart (real) ?1) (fun (fun (cart (real) ?1) (bool))
  (bool))) IN) (v (cart (real) ?1) w)) (v (fun (cart (real)
  ?1) (bool)) t))) (a (a (c (fun (cart (real) ?1) (fun (fun
```

```
(cart (real) ?1) (bool)) (bool))) IN) (v (cart (real) ?1)
w)) (a (c (fun (prod (cart (real) ?1) (real)) (fun (cart
(real) ?1) (bool))) ball) (a (a (c (fun (cart (real) ?1)
(fun (real) (prod (cart (real) ?1) (real)))) ,) (v (cart
(real) ?1) b)) (a (c (fun (num) (real)) real_of_num) (a (c
(fun (num) (num)) NUMERAL) (a (c (fun (num) (num)) BIT1)
(c (num) _0)))))))))))) (a (c (fun (fun (cart (real) ?1)
(bool)) (bool)) ?)  (l (v (cart (real) ?1) w) (a (a (c (fun
(bool) (fun (bool) (bool))) ∧) (a (a (c (fun (cart (real)
?1) (fun (fun (cart (real) ?1) (bool)) (bool))) IN) (v (cart
(real) ?1) w)) (v (fun (cart (real) ?1) (bool)) u))) (a (a
(c (fun (cart (real) ?1) (fun (fun (cart (real) ?1) (bool))
(bool))) IN) (v (cart (real) ?1) w)) (a (c (fun (prod (cart
(real) ?1) (real)) (fun (cart (real) ?1) (bool))) ball) (a
(a (c (fun (cart (real) ?1) (fun (real) (prod (cart (real)
?1) (real)))) ,) (v (cart (real) ?1) b)) (a (c (fun (num)
(real)) real_of_num) (a (c (fun (num) (num)) NUMERAL) (a
(c (fun (num) (num)) BIT1) (c (num) _0)))))))))))))) (a (c
(fun (fun ?0 (bool)) (bool)) !)  (l (v ?0 x) (a (a (c (fun
(bool) (fun (bool) (bool))) ==>) (a (a (c (fun ?0 (fun (fun
?0 (bool)) (bool))) IN) (v ?0 x)) (v (fun ?0 (bool)) d)))
(a (c (fun (bool) (bool)) ∼) (a (a (c (fun (cart (real)
?1) (fun (fun (cart (real) ?1) (bool)) (bool))) IN) (a (v
(fun ?0 (cart (real) ?1)) g) (v ?0 x))) (a (a (c (fun (fun
(cart (real) ?1) (bool)) (fun (fun (cart (real) ?1) (bool))
(fun (cart (real) ?1) (bool)))) UNION) (v (fun (cart (real)
?1) (bool)) t)) (v (fun (cart (real) ?1) (bool)) u)))))))))
(a (c (fun (bool) (bool)) ∼) (a (c (fun (fun (cart (real)
?1) (bool)) (bool)) ?)  (l (v (cart (real) ?1) b) (a (a (c
(fun (fun (cart (real) ?1) (bool)) (fun (fun (cart (real)
?1) (bool)) (bool))) SUBSET) (a (c (fun (prod (cart (real)
?1) (real)) (fun (cart (real) ?1) (bool))) ball) (a (a
(c (fun (cart (real) ?1) (fun (real) (prod (cart (real)
?1) (real)))) ,) (v (cart (real) ?1) b)) (a (c (fun (num)
(real)) real_of_num) (a (c (fun (num) (num)) NUMERAL) (a
(c (fun (num) (num)) BIT1) (c (num) _0)))))))) (a (a (c (fun
(fun ?0 (cart (real) ?1)) (fun (fun ?0 (bool)) (fun (cart
(real) ?1) (bool)))) IMAGE) (v (fun ?0 (cart (real) ?1)) g))
(v (fun ?0 (bool)) d)))))))))))))
```

Ground truth: `<START> (fun (fun (fun (cart (real) ?1) (bool)) (bool)) (bool)) <END>`

Source theorem pretty printed: `!t u.  (!b.  (?w.  w IN t ∧ w IN ball (b,&1)) ∨ (?w.  w IN u ∧ w IN ball (b,&1))) ∧ (!x.  x IN d ==> ∼(g x IN t UNION u)) ==> ∼(?b.  ball (b, &1) SUBSET IMAGE g d)`

- ```
  (<theorem> (a (c <PREDICT> !)  (l (v (fun (cart (real) N)
  (bool)) s) (a (a (c (fun (bool) (fun (bool) (bool))) =) (a
  (c (fun (fun (cart (real) N) (bool)) (bool)) is_interval)
  (a (a (c (fun (fun (cart (real) N) (cart (real) N)) (fun
  (fun (cart (real) N) (bool)) (fun (cart (real) N) (bool))))
  IMAGE) (c (fun (cart (real) N) (cart (real) N)) vector_neg))
  (v (fun (cart (real) N) (bool)) s)))) (a (c (fun (fun (cart
  (real) N) (bool)) (bool)) is_interval) (v (fun (cart (real)
  N) (bool)) s))))))
  ```

  Ground truth: `<START> (fun (fun (fun (cart (real) N) (bool)) (bool)) (bool)) <END>`

  Source theorem pretty printed: `!s.  is_interval (IMAGE (−) s) <=> is_interval s`

- (<theorem> (a (c (fun (fun (real) (bool)) (bool)) !)  (l (v
  (real) x) (a (a (a (c (fun (fun (real) (real)) (fun (real)
  (fun (net (real)) (bool)))) has_real_derivative) (c (fun
  (real) (real)) atn)) (a (c (fun (real) (real)) real_inv) (a
  (a (c (fun (real) (fun (real) (real))) real_add) (a (c (fun
  (num) (real)) real_of_num) (a (c (fun (num) (num)) NUMERAL)
  (a (c (fun (num) (num)) BIT1) (c (num) _0))))) (a (a (c (fun
  (real) (fun (num) (real))) real_pow) (v <PREDICT> x)) (a (c
  (fun (num) (num)) NUMERAL) (a (c (fun (num) (num)) BIT0) (a
  (c (fun (num) (num)) BIT1) (c (num) _0)))))))))) (a (c (fun
  (real) (net (real))) atreal) (v (real) x))))))

  Ground truth: <START> (real) <END>

  Source theorem pretty printed: !x.  (atn has_real_derivative inv (&1 +
  x pow 2)) (atreal x)

- (<theorem> (a (a (c (fun (fun ?0 (bool)) (fun (fun ?0
  (bool)) (bool))) =) (a (a (c (fun (fun ?0 (bool)) (fun (fun
  ?0 (bool)) (fun ?0 (bool)))) INTER) (v (fun ?0 (bool)) s))
  (a (a (c (fun (fun ?0 (bool)) (fun (fun ?0 (bool)) (fun ?0
  (bool)))) UNION) (v (fun ?0 (bool)) t)) (v (fun ?0 (bool))
  u)))) (a (a (c (fun (fun ?0 (bool)) (fun (fun ?0 (bool))
  (fun ?0 (bool)))) UNION) (a (a (c <PREDICT> INTER) (v (fun
  ?0 (bool)) s)) (v (fun ?0 (bool)) t))) (a (a (c (fun (fun
  ?0 (bool)) (fun (fun ?0 (bool)) (fun ?0 (bool)))) INTER) (v
  (fun ?0 (bool)) s)) (v (fun ?0 (bool)) u)))))

  Ground  truth:  <START> (fun (fun ?0 (bool)) (fun (fun ?0 (bool))
  (fun ?0 (bool)))) <END>

  Source theorem pretty printed: s INTER (t UNION u) = s INTER t UNION s
  INTER u

- (<theorem> (a (a (c (fun (real) (fun (real) (bool))) =) (a
  (c (fun (cart (real) ?0) (real)) infnorm) (a (c (fun (num)
  (cart (real) ?0)) vec) (a (c (fun (num) (num)) NUMERAL) (c
  (num) _0))))) (a (c (fun (num) (real)) real_of_num) (a (c
  (fun (num) (num)) NUMERAL) (c <PREDICT> _0)))))

  Ground truth: <START> (num) <END>

  Source theorem pretty printed: infnorm (vec 0) = &0

**Hard Type Inference.**

- (<theorem> (a (c <MASK> !)  (l (v <MASK> s) (a (a (c <MASK>
  =) (a (c <MASK> INTERS) (v <MASK> s))) (a (a (c <PREDICT>
  DIFF) (c <MASK> UNIV)) (a (c <MASK> UNIONS) (a (c <MASK>
  GSPEC) (l (v <MASK> GEN%PVAR%0) (a (c <MASK> ?)  (l (v
  <MASK> t) (a (a (a (c <MASK> SETSPEC) (v <MASK> GEN%PVAR%0))
  (a (a (c <MASK> IN) (v <MASK> t)) (v <MASK> s))) (a (a (c
  <MASK> DIFF) (c <MASK> UNIV)) (v <MASK> t)))))))))))))))

  Ground  truth:  <START> (fun (fun ?0 (bool)) (fun (fun ?0 (bool))
  (fun ?0 (bool)))) <END>

  Source theorem pretty printed: !s.  INTERS s = (:?0) DIFF UNIONS {(:?0)
  DIFF t | t IN s}

- (<theorem> (a (c <MASK> !)  (l (v <MASK> f) (a (c <MASK>
  !)  (l (v <MASK> s) (a (a (c <MASK> =) (a (a (c <MASK>
  uniformly_continuous_on) (v <MASK> f)) (v <MASK> s))) (a
  (c <MASK> !)  (l (v <MASK> e) (a (a (c <MASK> ==>) (a (a
  (c <MASK> real_lt) (a (c <MASK> real_of_num) (a (c <MASK>
  NUMERAL) (c <MASK> _0)))) (v <MASK> e))) (a (c <MASK> ?)  (l
  (v <MASK> d) (a (a (c <MASK> ∧) (a (a (c <MASK> real_lt)

```
(a (c <MASK> real_of_num) (a (c <MASK> NUMERAL) (c <MASK>
_0)))) (v <MASK> d))) (a (c <MASK> !)  (l (v <MASK> t) (a
(c <MASK> !)  (l (v <MASK> t') (a (a (c <MASK> ==>) (a (a
(c <MASK> ∧) (a (a (c <MASK> SUBSET) (v <MASK> t)) (v <MASK>
s))) (a (a (c <MASK> ∧) (a (a (c <MASK> SUBSET) (v <PREDICT>
t')) (v <MASK> s))) (a (a (c <MASK> ∧) (a (c <MASK> bounded)
(v <MASK> t))) (a (a (c <MASK> ∧) (a (c <MASK> bounded) (v
<MASK> t'))) (a (a (c <MASK> real_lt) (a (c <MASK> hausdist)
(a (a (c <MASK> ,) (v <MASK> t')) (v <MASK> t)))) (v <MASK>
d))))))) (a (a (c <MASK> real_lt) (a (c <MASK> hausdist)
(a (a (c <MASK> ,) (a (a (c <MASK> IMAGE) (v <MASK> f)) (v
<MASK> t'))) (a (a (c <MASK> IMAGE) (v <MASK> f)) (v <MASK>
t))))) (v <MASK> e)))))))))))))))))))
```
Ground truth: <START> (fun (cart (real) M) (bool)) <END>

Source theorem pretty printed:  !f s.  f uniformly_continuous_on s
<=> (!e.  &0 < e ==> (?d.  &0 < d ∧ (!t t'.  t SUBSET s ∧ t'
SUBSET s ∧ bounded t ∧ bounded t' ∧ hausdist (t',t) < d ==>
hausdist (IMAGE f t', IMAGE f t) < e)))

- ```
(<theorem> (a (a (c <MASK> ==>) (a (a (c <PREDICT> IN) (v
<MASK> a)) (v <MASK> s))) (a (a (c <MASK> =) (a (a (c <MASK>
DIFF) (a (a (c <MASK> INSERT) (v <MASK> a)) (a (a (c <MASK>
DELETE) (v <MASK> t)) (v <MASK> b)))) (v <MASK> s))) (a (a
(c <MASK> DELETE) (a (a (c <MASK> DIFF) (v <MASK> t)) (v
<MASK> s))) (v <MASK> b)))))
  ```
  Ground    truth:    <START> (fun ?0 (fun (fun ?0 (bool)) (bool)))
  <END>

  Source theorem pretty printed: a IN s ==> a INSERT (t DELETE b) DIFF s
  = (t DIFF s) DELETE b

- ```
(<theorem> (a (c <MASK> !)  (l (v <PREDICT> b) (a (c <MASK>
convex) (a (c <MASK> GSPEC) (l (v <MASK> GEN%PVAR%0) (a (c
<MASK> ?)  (l (v <MASK> z) (a (a (a (c <MASK> SETSPEC) (v
<MASK> GEN%PVAR%0)) (a (a (c <MASK> real_gt) (a (c <MASK>
Im) (v <MASK> z))) (v <MASK> b))) (v <MASK> z)))))))))))
  ```
  Ground truth: <START> (real) <END>

  Source theorem pretty printed: !b.   convex {z | Im z > b}

- ```
(<theorem> (a (c <MASK> !)  (l (v <MASK> x) (a (a (c <MASK>
==>) (a (c <MASK> ~) (a (a (c <MASK> nadd_eq) (v <MASK> x))
(a (c <MASK> nadd_of_num) (a (c <MASK> NUMERAL) (c <MASK>
_0)))))) (a (c <MASK> ?)  (l (v <MASK> B) (a (c <MASK> ?)
(l (v <MASK> N) (a (c <MASK> !)  (l (v <MASK> m) (a (c
<MASK> !)  (l (v <MASK> n) (a (a (c <MASK> ==>) (a (a (c
<MASK> ∧) (a (a (c <MASK> <=) (v <MASK> N)) (v <MASK> m)))
(a (a (c <MASK> <=) (v <MASK> N)) (v <MASK> n)))) (a (a
(c <MASK> <=) (a (a (c <MASK> *) (a (a (c <MASK> *) (a (a
(c <MASK> dest_nadd) (v <MASK> x)) (v <MASK> m))) (a (a (c
<MASK> dest_nadd) (v <MASK> x)) (v <MASK> n)))) (a (c <MASK>
dist) (a (a (c <MASK> ,) (a (a (c <MASK> *) (v <MASK> m)) (a
(a (c <MASK> nadd_rinv) (v <MASK> x)) (v <PREDICT> n)))) (a
(a (c <MASK> *) (v <MASK> n)) (a (a (c <MASK> nadd_rinv) (v
<MASK> x)) (v <MASK> m))))))) (a (a (c <MASK> *) (v <MASK>
B)) (a (a (c <MASK> *) (a (a (c <MASK> *) (v <MASK> m))
(v <MASK> n))) (a (a (c <MASK> +) (v <MASK> m)) (v <MASK>
n))))))))))))))))))))
  ```
  Ground truth: <START> (num) <END>

  Source theorem pretty printed: !x.   ~(x === &0) ==> (?B N. !m n.  N <=
  m ∧ N <= n ==> (fn x m * fn x n) * dist (m * nadd_rinv x n,n
  * nadd_rinv x m) <= B * (m * n) * (m + n))
```

**Assumptions.**

- Prompt: `(<theorem> (a (a (c (fun (bool) (fun (bool) (bool)))` `==>) (a (a (c (fun (fun ?1 (bool)) (fun (fun ?1 (bool))` `(bool))) =) (a (c (fun (fun ?1 (bool)) (fun ?1 (bool)))` `GSPEC) (l (v ?1 GEN%PVAR%0) (a (c (fun (fun ?1 (bool))` `(bool)) ?) (l (v ?1 x) (a (a (a (c (fun ?1 (fun (bool)` `(fun ?1 (bool)))) SETSPEC) (v ?1 GEN%PVAR%0)) (a (a (c (fun` `(bool) (fun (bool) (bool))) ∧) (a (a (c (fun ?1 (fun (fun` `?1 (bool)) (bool))) IN) (v ?1 x)) (v (fun ?1 (bool)) s))) (a` `(a (c (fun ?0 (fun ?0 (bool))) =) (a (v (fun ?1 ?0) f) (v ?1` `x))) (v ?0 a)))) (v ?1 x))))))) (v (fun ?1 (bool)) t))) (a` `(a (c (fun (bool) (fun (bool) (bool))) ==>) <PREDICT>) (a (c` `(fun (fun ?1 (bool)) (bool)) !) (l (v ?1 x) (a (a (c (fun` `(bool) (fun (bool) (bool))) ==>) (a (a (c (fun (bool) (fun` `(bool) (bool))) ∧) (a (v (fun ?1 (bool)) P) (v ?1 x))) (a (v` `(fun ?1 (bool)) Q) (v ?1 x)))) (a (c (fun (bool) (bool)) ∼)` `(a (a (c (fun ?0 (fun ?0 (bool))) =) (a (v (fun ?1 ?0) f) (v` `?1 x))) (v ?0 a)))))))))`

  Ground truth: `<START> (a (a (c (fun (bool) (fun (bool) (bool)))` `∧) (a (c (fun (fun ?1 (bool)) (bool)) !) (l (v ?1 x) (a` `(a (c (fun (bool) (fun (bool) (bool))) ==>) (a (v (fun ?1` `(bool)) P) (v ?1 x))) (a (a (c (fun ?1 (fun (fun ?1 (bool))` `(bool))) IN) (v ?1 x)) (v (fun ?1 (bool)) s)))))) (a (c (fun` `(fun ?1 (bool)) (bool)) !) (l (v ?1 x) (a (a (c (fun (bool)` `(fun (bool) (bool))) ==>) (a (a (c (fun (bool) (fun (bool)` `(bool))) ∧) (a (v (fun ?1 (bool)) P) (v ?1 x))) (a (v (fun` `?1 (bool)) Q) (v ?1 x)))) (a (c (fun (bool) (bool)) ∼) (a` `(a (c (fun ?1 (fun (fun ?1 (bool)) (bool))) IN) (v ?1 x)) (v` `(fun ?1 (bool)) t))))))) <END>`

  Source theorem pretty printed: `{x | x IN s ∧ f x = a} = t ==> (!x.  P` `x ==> x IN s) ∧ (!x.  P x ∧ Q x ==> ∼(x IN t)) ==> (!x.  P` `x ∧ Q x ==> ∼(f x = a))`

- Prompt: `(<theorem> (a (c (fun (fun (fun (cart (real) N)` `(bool)) (bool)) (bool)) !) (l (v (fun (cart (real) N)` `(bool)) s) (a (a (c (fun (bool) (fun (bool) (bool))) ==>)` `<PREDICT>) (a (a (c (fun (fun (cart (real) N) (bool)) (fun` `(fun (cart (real) N) (bool)) (bool))) =) (a (c (fun (fun` `(cart (real) N) (bool)) (fun (cart (real) N) (bool)))` `inside) (v (fun (cart (real) N) (bool)) s))) (c (fun (cart` `(real) N) (bool)) EMPTY))))))`

  Ground truth: `<START> (a (a (c (fun (bool) (fun (bool)` `(bool))) ∧) (a (c (fun (fun (cart (real) N) (bool)) (bool))` `connected) (a (a (c (fun (fun (cart (real) N) (bool)) (fun` `(fun (cart (real) N) (bool)) (fun (cart (real) N) (bool))))` `DIFF) (c (fun (cart (real) N) (bool)) UNIV)) (v (fun (cart` `(real) N) (bool)) s)))) (a (c (fun (bool) (bool)) ∼) (a (c` `(fun (fun (cart (real) N) (bool)) (bool)) bounded) (a (a (c` `(fun (fun (cart (real) N) (bool)) (fun (fun (cart (real) N)` `(bool)) (fun (cart (real) N) (bool)))) DIFF) (c (fun (cart` `(real) N) (bool)) UNIV)) (v (fun (cart (real) N) (bool))` `s))))) <END>`

  Source theorem pretty printed: `!s.  connected ((:realN) DIFF s) ∧` `∼bounded ((:realN) DIFF s) ==> inside s = {}`

- Prompt: `(<theorem> (a (a (c (fun (bool) (fun (bool) (bool)))` `==>) (a (a (c (fun (bool) (fun (bool) (bool))) ∧) (v (bool)` `q)) (a (c (fun (bool) (bool)) ∼) (v (bool) p)))) (a (a (c`

```
(fun (bool) (fun (bool) (bool))) ==>) <PREDICT>) (v (bool)
r))))
```

Ground truth: `<START> (a (a (c (fun (bool) (fun (bool) (bool))) =) (v (bool) p)) (v (bool) q)) <END>`

Source theorem pretty printed: `q ∧ ~p ==> (p <=> q) ==> r`

- Prompt: `(<theorem> (a (c (fun (fun (fun (cart (real) N) (real)) (bool)) (bool)) !) (l (v (fun (cart (real) N) (real)) f) (a (c (fun (fun (fun (real) (real)) (bool)) (bool)) !) (l (v (fun (real) (real)) g) (a (c (fun (fun (cart (real) N) (bool)) (bool)) !) (l (v (cart (real) N) x) (a (a (c (fun (bool) (fun (bool) (bool))) ==>) <PREDICT>) (a (a (c (fun (fun (cart (real) N) (real)) (fun (net (cart (real) N)) (bool))) real_continuous) (a (a (c (fun (fun (real) (real)) (fun (fun (cart (real) N) (real)) (fun (cart (real) N) (real)))) o) (v (fun (real) (real)) g)) (v (fun (cart (real) N) (real)) f))) (a (c (fun (cart (real) N) (net (cart (real) N))) at) (v (cart (real) N) x)))))))))))`

  Ground truth: `<START> (a (a (c (fun (bool) (fun (bool) (bool))) ∧) (a (a (c (fun (fun (cart (real) N) (real)) (fun (net (cart (real) N)) (bool))) real_continuous) (v (fun (cart (real) N) (real)) f)) (a (c (fun (cart (real) N) (net (cart (real) N))) at) (v (cart (real) N) x)))) (a (a (c (fun (fun (real) (real)) (fun (net (real)) (bool))) real_continuous) (v (fun (real) (real)) g)) (a (a (c (fun (net (real)) (fun (fun (real) (bool)) (net (real)))) within) (a (c (fun (real) (net (real))) atreal) (a (v (fun (cart (real) N) (real)) f) (v (cart (real) N) x)))) (a (a (c (fun (fun (cart (real) N) (real)) (fun (fun (cart (real) N) (bool)) (fun (real) (bool)))) IMAGE) (v (fun (cart (real) N) (real)) f)) (c (fun (cart (real) N) (bool)) UNIV))))) <END>`

  Source theorem pretty printed: `!f g x. f real_continuous at x ∧ g real_continuous atreal (f x) within IMAGE f (:realˆN) ==> g o f real_continuous at x`

- Prompt: `(<theorem> (a (c (fun (fun (fun (cart (real) M) (cart (real) N)) (bool)) (bool)) !) (l (v (fun (cart (real) M) (cart (real) N)) f) (a (c (fun (fun (fun (cart (real) M) (cart (real) P)) (bool)) (bool)) !) (l (v (fun (cart (real) M) (cart (real) P)) g) (a (c (fun (fun (fun (cart (real) M) (bool)) (bool)) (bool)) !) (l (v (fun (cart (real) M) (bool)) s) (a (c (fun (fun (num) (bool)) (bool)) !) (l (v (num) n) (a (a (c (fun (bool) (fun (bool) (bool))) ==>) <PREDICT>) (a (a (a (c (fun (num) (fun (fun (cart (real) M) (bool)) (fun (fun (cart (real) M) (cart (real) (finite_sum N P))) (bool)))) baire) (v (num) n)) (v (fun (cart (real) M) (bool)) s)) (l (v (cart (real) M) x) (a (a (c (fun (cart (real) N) (fun (cart (real) P) (cart (real) (finite_sum N P)))) pastecart) (a (v (fun (cart (real) M) (cart (real) N)) f) (v (cart (real) M) x))) (a (v (fun (cart (real) M) (cart (real) P)) g) (v (cart (real) M) x))))))))))))))`

  Ground truth: `<START> (a (a (c (fun (bool) (fun (bool) (bool))) ∧) (a (a (a (c (fun (num) (fun (fun (cart (real) M) (bool)) (fun (fun (cart (real) M) (cart (real) N)) (bool)))) baire) (v (num) n)) (v (fun (cart (real) M) (bool)) s)) (v (fun (cart (real) M) (cart (real) N)) f))) (a (a (a (c (fun (num) (fun (fun (cart (real) M) (bool)) (fun (fun (cart (real) M) (cart (real) P)) (bool)))) baire) (v (num) n)) (v (fun (cart`

```
(real) M) (bool)) s)) (v (fun (cart (real) M) (cart (real)
P)) g))) <END>
```

Source theorem pretty printed: `!f g s n.  baire n s f ∧ baire n s g ==>`
`baire n s (lambda x.  pastecart (f x) (g x))`

**Equalities.**

- Prompt: `(<theorem> (a (c (fun (fun (fun ?0 (cart (real) (2)))`
`(bool)) (bool)) !)  (l (v (fun ?0 (cart (real) (2))) f) (a`
`(c (fun (fun (fun ?0 (cart (real) (2))) (bool)) (bool)) !)`
`(l (v (fun ?0 (cart (real) (2))) g) (a (c (fun (fun (fun`
`?0 (bool)) (bool)) (bool)) !)  (l (v (fun ?0 (bool)) s) (a`
`(a (c (fun (bool) (fun (bool) (bool))) ==>) (a (c (fun (fun`
`?0 (bool)) (bool)) FINITE) (v (fun ?0 (bool)) s))) (a (a (c`
`(fun (cart (real) (2)) (fun (cart (real) (2)) (bool))) =) (a`
`(a (c (fun (fun ?0 (bool)) (fun (fun ?0 (cart (real) (2)))`
`(cart (real) (2)))) cproduct) (v (fun ?0 (bool)) s)) (l (v`
`?0 x) (a (a (c (fun (cart (real) (2)) (fun (cart (real) (2))`
`(cart (real) (2)))) complex_mul) (a (v (fun ?0 (cart (real)`
`(2))) f) (v ?0 x))) (a (v (fun ?0 (cart (real) (2))) g) (v`
`?0 x)))))) <PREDICT>)))))))))`

Ground truth: `<START> (a (a (c (fun (cart (real) (2)) (fun (cart`
`(real) (2)) (cart (real) (2)))) complex_mul) (a (a (c (fun`
`(fun ?0 (bool)) (fun (fun ?0 (cart (real) (2))) (cart (real)`
`(2)))) cproduct) (v (fun ?0 (bool)) s)) (v (fun ?0 (cart`
`(real) (2))) f))) (a (a (c (fun (fun ?0 (bool)) (fun (fun ?0`
`(cart (real) (2))) (cart (real) (2)))) cproduct) (v (fun ?0`
`(bool)) s)) (v (fun ?0 (cart (real) (2))) g))) <END>`

Source theorem pretty printed: `!f g s.  FINITE s ==> cproduct s (\x.  f`
`x * g x) = cproduct s f * cproduct s g`

- Prompt:         `(<theorem> (a (c (fun (fun (fun (cart (real) N)`
`(bool)) (bool)) (bool)) !)  (l (v (fun (cart (real) N)`
`(bool)) s) (a (c (fun (fun (fun (cart (real) N) (bool))`
`(bool)) (bool)) !)  (l (v (fun (cart (real) N) (bool)) t)`
`(a (a (c (fun (bool) (fun (bool) (bool))) ==>) (a (a (c (fun`
`(bool) (fun (bool) (bool))) ∧) (a (c (fun (fun (cart (real)`
`N) (bool)) (bool)) convex) (v (fun (cart (real) N) (bool))`
`s))) (a (a (c (fun (bool) (fun (bool) (bool))) ∧) (a (c (fun`
`(fun (cart (real) N) (bool)) (bool)) affine) (v (fun (cart`
`(real) N) (bool)) t))) (a (c (fun (bool) (bool)) ~) (a (a`
`(c (fun (fun (cart (real) N) (bool)) (fun (fun (cart (real)`
`N) (bool)) (bool))) =) (a (a (c (fun (fun (cart (real) N)`
`(bool)) (fun (fun (cart (real) N) (bool)) (fun (cart (real)`
`N) (bool)))) INTER) (a (c (fun (fun (cart (real) N) (bool))`
`(fun (cart (real) N) (bool))) relative_interior) (v (fun`
`(cart (real) N) (bool)) s))) (v (fun (cart (real) N) (bool))`
`t))) (c (fun (cart (real) N) (bool)) EMPTY)))))) (a (a (c`
`(fun (fun (cart (real) N) (bool)) (fun (fun (cart (real)`
`N) (bool)) (bool))) =) <PREDICT>) (a (a (c (fun (fun (cart`
`(real) N) (bool)) (fun (fun (cart (real) N) (bool)) (fun`
`(cart (real) N) (bool)))) INTER) (a (c (fun (fun (cart`
`(real) N) (bool)) (fun (cart (real) N) (bool))) closure)`
`(v (fun (cart (real) N) (bool)) s))) (v (fun (cart (real) N)`
`(bool)) t)))))))))`

Ground truth: `<START> (a (c (fun (fun (cart (real) N) (bool))`
`(fun (cart (real) N) (bool))) closure) (a (a (c (fun (fun`
`(cart (real) N) (bool)) (fun (fun (cart (real) N) (bool))`

```
(fun (cart (real) N) (bool)))) INTER) (v (fun (cart (real)
N) (bool)) s)) (v (fun (cart (real) N) (bool)) t))) <END>
```
Source theorem pretty printed: `!s t. convex s ∧ affine t ∧ ~(relative_interior s INTER t = {}) ==> closure (s INTER t) = closure s INTER t`

- Prompt: `(<theorem> (a (a (c (fun (bool) (fun (bool) (bool))) ==>) (a (a (c (fun (fun ?0 (bool)) (fun (fun ?0 (bool)) (bool))) SUBSET) (v (fun ?0 (bool)) t)) (a (a (c (fun (fun ?0 (bool)) (fun (fun ?0 (bool)) (fun ?0 (bool)))) DIFF) (c (fun ?0 (bool)) UNIV)) (v (fun ?0 (bool)) s)))) (a (a (c (fun (fun ?0 (bool)) (fun (fun ?0 (bool)) (bool))) =) <PREDICT>) (c (fun ?0 (bool)) EMPTY))))`

  Ground truth: `<START> (a (a (c (fun (fun ?0 (bool)) (fun (fun ?0 (bool)) (fun ?0 (bool)))) INTER) (v (fun ?0 (bool)) s)) (v (fun ?0 (bool)) t)) <END>`

  Source theorem pretty printed: `t SUBSET (:?0) DIFF s ==> s INTER t = {}`

- Prompt: `(<theorem> (a (c (fun (fun (real) (bool)) (bool)) !) (l (v (real) x) (a (a (c (fun (real) (fun (real) (bool))) =) <PREDICT>) (a (c (fun (real) (real)) real_abs) (v (real) x))))))`

  Ground truth: `<START> (a (a (c (fun (real) (fun (num) (real))) real_pow) (a (c (fun (real) (real)) sqrt) (v (real) x))) (a (c (fun (num) (num)) NUMERAL) (a (c (fun (num) (num)) BIT0) (a (c (fun (num) (num)) BIT1) (c (num) _0))))) <END>`

  Source theorem pretty printed: `!x. sqrt x pow 2 = abs x`

- Prompt: `(<theorem> (a (a (c (fun (fun A (bool)) (fun (fun A (bool)) (bool))) =) <PREDICT>) (a (c (fun (fun A (bool)) (fun A (bool))) GSPEC) (l (v A GEN%PVAR%0) (a (c (fun (fun A (bool)) (bool)) ?) (l (v A y) (a (a (a (c (fun A (fun (bool) (fun A (bool)))) SETSPEC) (v A GEN%PVAR%0)) (a (a (c (fun (bool) (fun (bool) (bool))) ∨) (a (a (c (fun A (fun (fun A (bool)) (bool))) IN) (v A y)) (v (fun A (bool)) s))) (a (a (c (fun A (fun A (bool))) =) (v A y)) (v A x)))) (v A y)))))))`

  Ground truth: `<START> (a (a (c (fun A (fun (fun A (bool)) (fun A (bool)))) INSERT) (v A x)) (v (fun A (bool)) s)) <END>`

  Source theorem pretty printed: `x INSERT s = {y | y IN s ∨ y = x}`

