# OpenReview forum: "Mathematical Reasoning via Self-supervised Skip-tree Training"
_ICLR.cc/2021/Conference — ICLR 2021 Spotlight_

### Official Review · AnonReviewer2 · 2020-10-26
**Mathematical Reasoning via Self-supervised Skip-tree Training**

**Rating:** 7
**Confidence:** 4

**Review:**

The authors propose a self-supervised learning task to enhance the reasoning capabilities of machine learning models on mathematical formulas and to perform conjecturing in higher order logic. The task consists in masking out specific portions of mathematical statements and predict them from the surrounding parts. The task can (i) be used during training, to provide supervisory signal to the machine learning model and to increase the effective size of the otherwise small training dataset, and (ii) be used during testing, to evaluate the reasoning capabilities of the learnt models by masking out the mathematical statements at different level of granularities. The authors perform an extensive experimental analysis and provide evidence on the utility of using self-supervised learning in the context of theorem proving.

Overall, the paper is clearly written and the bibliography complete. Also, the experimental analysis is undoubtedly valuable for the machine learning community. In fact, it provides additional empirical evidence to the works of [1-2], in the sense that self-supervised learning can be used not only as a pre-training stage for the machine learning models, but also as a task to perform conjecturing.

There are major issues though. In particular, there are issues in terms of the originality of the proposed task and the reproducibility of the experiments, thus obscuring the positive aspects of the paper. Please, see below for more detailed comments.

In lieu of this, I consider the paper marginally below the acceptance threshold and therefore recommend for an initial rejection. Nevertheless, I'm willing to raise my score if the authors properly address the issues highlighted below.

DETAILED COMMENTS

Originality: The proposed task seems to be identical to the task in [3], except for the fact that the task operates on mathematical statements rather than on natural language ones. Importantly, both kinds of sentences/statements share a tree-structured representation, which makes the task in [3] trivially applicable to the mathematical context. Can the authors highlight other differences or explain if this is not correct?

Reproducibility: First of all, it's extremely important to add in the section about results and discussion detailed information about the computational resources and the time required for training, as well as the size of the machine learning model used in the experiments. In fact, it is well-known from other domains, like vision and natural language processing, that the performance of self-supervised learning models increase proportionally with the size of the model, at the expense of the training computational resources.
Furthermore, it's important to discuss about the methodology used to choose the hyperparameters. Appendix A only states that "we explored encoders and decoders with up to 12 layers and various learning rates and intermediate sizes."
Finally, do the authors plan to release the code?

FURTHER IMPORTANT COMMENTS

Answering to this following comments could be helpful to clarify about the originality and the novelty of the proposed task.

Distinguishing between training and evaluation tasks and saying that "several tasks" are proposed (i.e. in abstract, introduction, section 5 and conclusions) can be misleading for the reader, because he/she could think that they are different. Nevertheless, these tasks are all equivalent to the main skip-tree task, differing only in the way the terms are masked. Is that correct? If so, I would suggest to rephrase by saying that the main skip-tree task is quite flexible at masking statements. In fact, depending on which portion of statement is masked, the model can be induced to perform reasoning at different level of abstraction. Based on this, experiments are categorised according to the different ways of masking.

Furthermore, it is not completely clear when the authors says that "most previous works...have focused...in supervised training settings. In contrast, we train language models on an unsupervised proxy task" (related work, but also introduction).
Is the proposed training unsupervised? First of all, the data used in training contains only valid theorems. This can be already considered as a supervised information, because collecting valid theorems requires human effort. Secondly, the evaluation tasks are essentially similar to the training task and therefore I would consider this more as a supervised learning strategy. Could you please elaborate more on this, also in the text?

MINOR COMMENTS

In the experiments about conjecturing (2nd paragraph), can you provide more details on how you generated the free-form conjectures?

Furthermore, how is novelty of generated statements defined? Consider the following examples for conjecturing on assumptions and free-form conjecturing.

Conjecturing from assumptions

Imagine you have

a+b=0 implies a=-b

and the task is given by

<PREDICT> implies a=-b

Is the statement 'a+b=c and c=0' considered new according to your definition?

Free-form conjecturing

Imagine you have

<THEOREM> for all x, x=x

and the task is given by

<THEOREM> <PREDICT>

Is the theorem 'for all y, y=y' considered new according to your definition?


[1] Li et al. Modelling High-Level Mathematical Reasoning in Mechanised Declarative Proofs. arXiv 2020

[2] Polu and Sutskever. Generative Language Modelling for Automated Theorem Provers. arXiv 2020

[3] Zhang et al. PEGASUS: Pre-training with Extracted Gap-sentences for Abstractive Summarization. ICML 2020

########################################

UPDATE

The paper provides important and novel empirical observations about the use of machine learning to perform mathematical reasoning. The authors have addressed and clarified some doubts about the originality of their idea and the reproducibility of their experiments in the discussion phase. I believe that the paper is now ready for publication and I'm happy to recommend for its acceptance.

---

> ### Author Response · Authors · 2020-11-17
> **Author response**
>
> Thanks for the detailed review!
>
> Originality:
> Pegasus [2] is a task defined on natural language and relies on the notion of sentences, while our approach relies on the tree structure of expressions. Pegasus only considers “whole sentences” [2, page 2, paragraph 4], so it does not consider tree structure. Additionally, our training task can create a much wider variety of training examples than PEGASUS as the number of subtrees is closer to the number of tokens. (We exploit that when we sample many training examples from each mathematical statement.) We actually think that the entire development of training tasks from MASS/UNILM (predict subsequences) to T5 (predict multiple subsequences) to PEGASUS (predict sentences) and finally to this work (predict subtrees) were all incremental steps when only examined for their technical merits. But their impact is undeniably large.
>
> The main novelty of this paper is perhaps more in the conceptual domain as it provides a new perspective on how to achieve mathematical reasoning abilities without supervised training data (as also AnonReviewer1 notes).
>
> Reproducibility:
> We added a description of the computational resources, the size of the models, and our evaluation methodology.
> Larger models seem to perform better indeed. We have preliminary results that suggest this (for a model with 110M parameters) and are happy to add them to the paper, if the reviewers agree that this would improve the paper.
>
> Code:
> We will release the code for training data generation.
>
> Choice of hyperparameters:
> Because each training run is relatively costly, we did not perform a full grid search for the hyperparameters. Instead we consulted the literature and expert opinions and tested just a couple of hyperparameters. For the skip-sequence models we tried a more extensive list of hyperparameters, as we wanted to make sure that the negative results here are stable under different choices of hyperparameters.
>
> Difference between training task and evaluation tasks:
> The evaluation tasks and the training task indeed share the same format, i.e. predict a part of a larger expression. This is intentional as it allows to test the model without fine-tuning on reasoning tasks. The distributions of the training and evaluation tasks are, in fact, disjoint, because the evaluation tasks are derived from different source data, the validation/test theorems. We improved the description of the relationship between training and evaluation tasks in the beginning of Section 5.
>
> Unsupervised vs self-supervised:
> We corrected the occurrence of “unsupervised” to “self-supervised”, which is a better term.
>
> On the question whether our technique qualifies as being unsupervised/self-supervised:
> It is a great question where the line between supervised and self-supervised techniques is exactly. We believe that our use of “self-supervised” is in line with other papers: Self-supervised translation techniques use monolingual corpora, which, with the same argument that reviewer 2 provided, could be seen as “supervised information”. We think that the use of arbitrary subexpressions as prediction targets is general enough to be applied to arbitrary sources. There is also nothing that keeps us from applying skip-tree training to statements that are not true. In one sense, in which we are clearly self-supervised is that the training data preparation does not require human labelers to produce labels.
>
> Generation of conjectures:
> We improved the description of the free-form conjecturing task. The procedure is quite simple: We simply ask the model to produce theorems by presenting the input sequence (<theorem>  <PREDICT>). So, the subexpression the models has to fill in must be an entire mathematical statement, and the <theorem> token (hopefully) makes the model produce a statement that could be a theorem (as opposed to the <goal> tag, which indicates intermediate statements of proofs). Given only this single input sequence, we use a beam search with beam width 1024 to produce enough outputs for a meaningful evaluation.
>
> How we measure the novelty of terms:
> We consider strict syntactic (string) equality.

---

> > ### Comment · AnonReviewer2 · 2020-11-21
> > **Reviewer Response**
> >
> > Thank you for your clarifications.
> >
> > I have no further comments about the originality of the proposed skip tree task, as I agree with you that it brings a significant improvement over previous training strategies (MASS/UNILM, T5 etc...), despite being of low technical novelty.
> >
> > Regarding experimental reproducibility, would it be possible to give more details about the computational resources (teraFLOPs of the 8x8 TPU, the amount/type of CPUs) and the time required to train and evaluate your model? I think that these details becomes extremely relevant for others who would like to replicate your results, especially because you are planning to release only part of your code.
> >
> > Concerning the difference between training and evaluation tasks, I appreciate the modification at the beginning of Section 5. If I understand correctly, the difference is only in using different data to derive the training and the evaluation tasks. It seems that there is no real novelty in distinguishing between them. Can you please merge contributions 2 and 3 in a single point in the introduction?

---

> > > ### Author Response · Authors · 2020-11-23
> > > **Author response**
> > >
> > > > Regarding experimental reproducibility, would it be possible to give more details about the computational resources (teraFLOPs of the 8x8 TPU, the amount/type of CPUs) and the time required to train and evaluate your model? I think that these details becomes extremely relevant for others who would like to replicate your results, especially because you are planning to release only part of your code.
> > >
> > > Each TPU (v3) has around 420 TeraFLOPS. The training runs took between 5 and 15 hours (depending on the length of the output  sequence). So, theoretically, each training run used between 1.4 and 4.2 Petaflop days. However, this kind of measurement may obviously overestimate the operations actually needed. The CPUs were used through a cloud service and they are a mix of recent Intel/AMD processors. We added more details on the used compute resources to the paper.
> > >
> > > Edit: there was a miscommunication on our side. The training code is already public and will be linked in the final version. See also the comment below.
> > >
> > > > Concerning the difference between training and evaluation tasks, I appreciate the modification at the beginning of Section 5. If I understand correctly, the difference is only in using different data to derive the training and the evaluation tasks. It seems that there is no real novelty in distinguishing between them. Can you please merge contributions 2 and 3 in a single point in the introduction?
> > >
> > > The evaluation tasks differ from skip-tree training the source data they are derived from and also the choice of which exact subexpression has to be predicted. While skip-tree training selects any subexpression, the evaluation tasks select very specific subexpressions. They only share the format, i.e. to predict a missing subexpression.
> > >
> > > We’d like to compare this to GPT-3-like pretraining and question-answering benchmarks: They share the same format, i.e. to complete a partially given text, but question-answering benchmarks are selected according to a very specific distribution (show the question, expect an answer).
> > >
> > > We followed the reviewers suggestion to merge the contributions into a single point, where we say that “We also introduce several evaluation tasks that are subsumed by skip-tree training (i.e. predict a missing subexpression), but test specific logical reasoning abilities to make the performance of the models interpretable.”

---

> > > > ### Author Response · Authors · 2020-11-23
> > > > **Remark on training code**
> > > >
> > > > Our whole model definition and training code is already in a public github, we will put a link in the final version (we omit the link here to preserve anonymity)
> > > >
> > > > We have not published the training-data generation code, but we consider placing the code there as well for reference.
> > > >
> > > > The code misses a few details specific to our environment, but it is relatively straightforward to make it run.

---

### Official Review · AnonReviewer1 · 2020-10-28
**Interesting direction/experiments. Could be more clear, however.**

**Rating:** 7
**Confidence:** 4

**Review:**

Quality

The paper is quite well written and the experiments seem well thought out. Please see specific comments below.

Clarity

The S-expression needs more explanation for the paper to be self-contained. What is 'A' in (v A x)?


For those of us who might not be very familiar with the datasets, it might be helpful if the paper to demonstrate more examples as to what these proofs are translated to, in mathematical notation or textual explanations. There are a few snippets here and there but I don't think I get a good grasp of what kinds of proofs we deal with. For example, the theorem examples in Appendix section D seem quite lengthy. With effort it should be decoded into normal text by readers, but it would be more convenient to demonstrate it directly.


In addition, the free-form conjecturing evaluation can be more clear. In this section, it doesn't entirely explain in details what makes the generated statements 'new'. Is it only exact match with the training set? If that's the case, how robust is exact matching on measuring the novelty?

On the usefulness of the conjectures, there's not much explanation on the RL experiments + DeepHOL theorem prover. What is it supposed to do / how robust does it measure usability?


Question: a beam search of 1024 seems rather large. How does the result look if no beam search is used?

Originality

	The introduced method 'skip-tree' is not that different from the usual self-supervised training techniques used to train language models. This feature does not contribute to the novelty that much. However, this approach is a new take on learning mathematical reasoning with self-supervised learning, rather than supervised learning like in other previous work.



Significance

	Deep learning for mathematical proofs is an interesting direction and is relatively unexplored, compared to other application areas.




High-Level Pros & Cons

Pros
	- The claim / conclusion for this work that self-supervised training can lead to mathematical reasoning is rather intriguing.
	- The proposed skip-tree technique seems to make a lot of difference for training.
	- Decent ablation study (not using <MASK> token, skip-sequence instead of skip-tree).
	- The evaluation tasks introduced seem interesting.

Cons
	- I am not 100% convinced that the ``mathematical reasoning'' demonstrated is beyond pattern recognition. I understand that the evaluation tasks are evaluated on validation set theorems which are not seen during training. However, how can we be sure that this is mathematical reasoning versus the model recognizing similar patterns (not necessarily exact match)

More explanations on this would be appreciated.
	- Low to moderate novelty for the skip-tree technique.


Overall, I learned a lot from this paper and I believe others can benefit from it as well.

---

> ### Author Response · Authors · 2020-11-17
> **Author response**
>
> We thank the reviewer for the insightful review.
>
> > S-expressions: What is 'A' in (v A x)?
>
> ‘A’ is a generic type variable. The expression (v A x) is a variable named x that could still have any type. We expanded the description of S-expressions.
>
> > the theorem examples in Appendix section D seem quite lengthy. With effort it should be decoded into normal text by readers, but it would be more convenient to demonstrate it directly.
>
> For the missing assumptions and equality completion tasks, we had already provided human readable versions of the ground-truth, but somehow we forgot to add them for the type inference and hard type inference tasks. This is now fixed.
>
> > Is it only an exact match with the training set?
>
> Yes, we only look for exact matches. We updated the formulation to make this clear.
>
> > What is it supposed to do / how robust does it measure [usefulness]?
>
> We count a conjectured theorem as useful only if it is picked up as a premise by the DeepHOL theorem prover. This measurement only says that for some proof this premise was necessary (DeepHOL includes a proof pruning step that, for each premise, tests if dropping the premise was still a valid proof). There may still be alternative proofs that do not require this particular premise, and even if a conjecture is not used as a premise, there might be theorems for which it would be very useful, but we may not have those in our list of tasks. So this should only be seen as a rough approximation of usefulness. We believe that the take-away is that this method managed to produce at least some new and useful theorems.
>
> > Question: a beam search of 1024 seems rather large. How does the result look if no beam search is used?
>
> We used beam width 1024 only for the conjecturing experiments where we wanted to generate many different expressions. For the other evaluations we use a beam search with width 8. If we only generate a single trace, our best models reach 25% to 30% on the missing assumptions and equality completion tasks.
>
> > I understand that the evaluation tasks are evaluated on validation set theorems which are not seen during training. However, how can we be sure that this is mathematical reasoning versus the model recognizing similar patterns (not necessarily exact match)
>
> We are not sure this is a “con”. We believe that our experiments show that, at the very least, “pattern recognition” can effectively simulate some forms of mathematical reasoning. We hope that this will contribute to the wider discussions in the community and help to understand the fundamental question of the relationship between reasoning and “pattern recognition”.

---

### Official Review · AnonReviewer3 · 2020-10-28
**Review of "Mathematical Reasoning via Self-supervised Skip-tree Training"**

**Rating:** 7
**Confidence:** 3

**Review:**

--------------------------------------------------------------------------------------------------------------------------------
Summary:

This paper proposes a skip-tree training task. The authors show that self-supervised language models (the Transformer architecture to be exact) trained on the proposed skip-tree training task for mathematic theorem proving enable mathematical reasoning capabilities. Moreover, no fine-tuning is required to achieve the reported reasoning capabilities. They compare the mathematical reasoning abilities of the skip-tree training task with skip-sequence and show an impressive performance improvement. Another interesting result is studying whether any useful (novel) conjectures can be generated by the model.

--------------------------------------------------------------------------------------------------------------------------------
Overall assessment:

I really enjoyed reading this paper. The authors work on an interesting problem that has been gaining popularity in the last few years. I believe that this problem domain is a suitable test bed to investigate the reasoning capabilities of language models. The authors propose a skip-tree training task and show that it significantly improves the performance of skip-sequence training. The idea is simple and nice: instead of masking out arbitrary sub-sequences (which is the case for skip-sequence), the authors propose to mask out sub-trees. The authors provide extensive ablation studies and evaluations to justify several design choices and to show the reasoning abilities of their model. I do have some comments about presentation clarity (look below), that I hope the authors can address during the rebuttal. Moreover, I am a bit skeptical about the almost 0 performance of the skip-sequence task under some scenarios. I will clarify this as well in the cons section below.

--------------------------------------------------------------------------------------------------------------------------------
Pros:

 (1) I like the idea of skip-tree training for structured problems. It seems to improve the performance of the model dramatically (from almost not being able to predict AT ALL to near perfect prediction — in the Hard inference problem for example). I do have a comment about this in the cons section.

 (2) The authors design a set of mathematical reasoning evaluations that are used as down-stream tasks (without fine-tuning) to evaluate the self-supervised trained models. They show good performance on all the tasks compared to their baselines.

 (3) The authors provide a set of ablation studies to justify their design choices.

 (4) I liked the conjecture experiment that evaluated whether the model can generate novel provable conjectures.

--------------------------------------------------------------------------------------------------------------------------------
Cons:

  (1) The performance of skip-sequence training on three out of the four evaluation tasks in Table 2 (Hard type inference, Assumptions, Equalities) are extremely poor. It is my understanding that the only difference between these models and the proposed skip-tree model is in the held-out sub-expression (I am not sure if the skip-sequence models have the <MASK> tokens in addition to <PREDICT> or not). If that understanding is correct, then this might indicate that almost all of the masked <PREDICT> sub-sequences are not “proper” trees. Is that correct? If so, does enforcing a variable portion (say from 0% to 100%) of the held-out sub-sequences to be sub-trees smoothly move the success rates reported in Table 2 towards the skip-tree’s performance?

  (1-1) Another possibility for the poor performance could be that the skip-sequence models were not trained with <MASK> tokens (I am just judging this based on the large difference between the skip-tree and the skip-tree (no mask) model). If this is the case, a fair comparison would also include the skip-sequence training with the additional <MASK> tokens. This allows the reader to understand which parts of the model result in the reported gains.

  (2) In figure 1 the authors illustrate the portion of the data that they use in their experiments for train and validation. It seems like the gray areas (specifically the test set) are not considered in the paper. What is confusing is that in Section 3 they do mention a train/validation/test split. But all the evaluation tasks defined in Section 5 in the paper seem to only refer to the validation data. There seems to be no indication of the use of a test dataset for the reported results (in Tables 2 and 3), which I find troubling. Especially because the authors do mention that they hyper-parameter optimized their models in Appendix A (not mentioned using which part of the data). Can you please clarify this (both in the rebuttal response and in the paper)?

  (3) How is a correct prediction assessed? It is mentioned that the model’s performance is evaluated for “exact match”. However, as mentioned in the Assumptions task, predicting y=x for the ground truth x=y should be counted as a “correct prediction”. If such predictions are not being counted as “correct”, I highly encourage the authors to add an evaluation metric that considers mathematical equivalence of the generated predictions and the ground-truth. A symbolic solver like Sympy or Mathematica might be able to at least verify the equivalence of simpler expressions (I am not sure how complex these expressions are).

  (3-1) To follow up on point (3) above, the authors mention that “to make a correct prediction our language models thus have to understand which statements are more general and also know about naming conventions”. I strongly disagree that this is a good way of measuring mathematical reasoning abilities. On the contrary, I think a model that has truly learned to mathematically reason, is one that ignores irrelevant details such as naming conventions or “generality” of the statement (generality is subjectively used by the authors and I think references to it should be removed from the paper).

  (4) It is not clearly stated what the authors mean by “new” in section 6.1. Are variations of already seen statements considered new? (e.g. if x=y is in the training set, would y=x be a new statement?). I recommend the authors clearly define this in the paper.

  (5) It is not clear to me what the takeaway is for the sampling strategy. In the main text it was implied that the weighted sampling will be better because it allows one to choose non-leafs more often. However, the results shown in Tables 2 and 3 show that some scenarios seem to be better with the uniform sampling and some with the weighted sampling. Do the authors have any comments/discussion on that?

--------------------------------------------------------------------------------------------------------------------------------
Smaller details:

  (1) Please use the correct citation command for references that are at the beginning of the sentence (e.g. section 2, Paragraph 2, line 4: Zhang et al. (2019)).

  (2) The authors mention that Lample and Charton (2020) “requires that the inverse of the prediction task can be computed”. Can you explain what this means?

  (3) The explanation of the s-expression given in Section 3 lacks enough details. I wasn’t able to fully understand the representation. Perhaps an illustration with step by step labels can make it easier to understand.

  (4) Section 3, Last paragraph: What does “the split is defined on the theorems” mean? In general this paragraph was somewhat hard to follow.

  (5) What portion of the training data is omitted as a result of what is described in Section 4, Paragraph 2.

  (6) I find the first argument about the reason for adding <MASK> tokens somewhat subjective. Why does making the task harder result in better performance?

  (7) I did not understand the multiple sample per dataset generation at all. Do the authors mean that for each math statement they generate n=100 samples with the <PREDICT> and <MASK> tokens? If so, this implies that there will be 360k*100 examples in the larger training set. But this is not consistent with the data stats provided in Table 1.

  (8) I think it would be very useful if the authors add human-readable equivalents of the s-expressions presented in Appendix D to that Appendix.

  (9) Section 6: What does 1M “steps” refer to? Is that the number of model updates?

  (10) It would be great to add a few sentences about what the reinforcement learning experiments in the DeepHOL prover are to make the paper more self-contained.

--------------------------------------------------------------------------------------------------------------------------------
Typos:

Sec 1, Parag 3: which are capable (to) of generating …
Sec 2, Parag 4: models to logic(s)

---

> ### Author Response · Authors · 2020-11-17
> **Author response to the major comments**
>
> Thanks a lot for the detailed comments!
>
> We first address the question in the "Cons" and then proceed with the minor remarks.
>
> > (1)  ... almost all of the masked <PREDICT> sub-sequences are not “proper” trees. Is that correct?
>
> Correct.
>
> > does enforcing a variable portion (say from 0% to 100%) of the held-out sub-sequences to be sub-trees smoothly move the success rates reported in Table 2 [...]?
>
> This sounds very likely. We started an experiment where we mixed skip-sequence and skip-tree training examples at a 1:1 ratio. We will report on the results in a couple of days.
>
> > (1-1) Another possibility for the poor performance could be that the skip-sequence models were not trained with <MASK> tokens
>
> The skip-sequence tasks indeed do not have the <MASK> token. This should not put them at a disadvantage for the type inference, missing assumptions, and equality completion tasks, as they don’t contain the <MASK> token either. For the hard type inference task we tried to make clear that these numbers are not comparable by graying them out in Table 2. Addressing this issue by masking out further subsequences with a <MASK> token in the skip-sequence tasks might be possible, but we do not see a reason to expect that the performance would surpass the performance on the regular type inference task, i.e. it is bounded by 45%.
>
> (2) train/validation/test split: The observation is correct, we did not use the test set. We had only tested a handful of  hyperparameters, as each experiment is somewhat expensive. But we agree that this is not optimal. We now reran the evaluation on the test set for the main experiment (skip-tree, weighted). There are no significant changes in the outcome. We will update the paper with the new numbers in the next few days when the experiments for the other models are complete.
>
> (3) We only count exact syntactic matches; i.e. “y=x” is not the same as “x=y”. This may indeed lead to an underestimation of the performance of the models (as also suggested by the experiments in Table 3). It would be nice to have a more semantical test, but this is harder than it may seem at first. We see three main issues with more semantic checks: First, Sympy and Mathematica (or some neural theorem prover) could only ever prove a portion of unknown size of these equalities and so our measurement of the language models would depend on how good these external algorithms are. Second, Sympy and Mathematica have no support for higher-order logic and we do not expect them to work particularly well. At the very least, their performance would depend highly on the quality of the translation into a compatible format. Third and last, for each evaluation task, the exact semantical test would need to differ: for type inference and equality completion, the semantic equivalence of the prediction and the ground truth is not the right criterion. To keep the measurements interpretable and reproducible, we therefore prefer to restrict the measurement in Table 2 to syntactic equivalence.
>
> (3-1)  This paragraph was supposed to shed light on other aspects of mathematics that the models have to learn besides the main reasoning task. We improved the wording.
>
> (4) Throughout the paper we consider syntactic equality. We slightly changed the final sentence in the paragraph “How often are predictions true and new?” to emphasize that we consider syntactic matches.
>
> (5) There is not a clear winner here. We adapted the wording in Section 4 to avoid setting other expectations. Originally we expected this to make a difference, but it turned out not to be that significant.

---

> > ### Author Response · Authors · 2020-11-17
> > **Addendum on <MASK> tokens and the skip-sequence task**
> >
> > After an internal discussion, we realized that there is a better angle to to understand the impact of the skip-tree vs skip-sequence choice. We have to compare the skip-sequence models to the model called "Skip-tree (no <MASK>)", which is also trained without the <MASK> tokens. This allows for an apples-to-apples comparison.
> >
> > We expanded the discussion of the results in the beginning of Section 6 to clarify this point. We will upload the updates together with the updated Table 2 (as we are rerunning the experiments with an evaluation on the test set).

---

> ### Author Response · Authors · 2020-11-17
> **Author response on "smaller details"**
>
> Also thanks for the smaller remarks. We have addressed those and answer the direct questions below:
>
> (2) Lample and Charton need to be able to compute the derivative of a formula and then predict the inverse of that operation (i.e. the integral). To transfer this style of training to other mathematical tasks, we need that one direction of the computation can be done algorithmically.
>
> (3) We expanded the description of s-expressions.
>
> (4) “What does ‘the split is defined on the theorems’ mean?” First, the theorems are split into train/valid/test, then the entire proof of each theorem is assigned into the same category as its theorem. We extended the description in the paper.
>
> (5) For the skip-tree (weighted) training dataset, less than 0.1% of the generated examples were dropped because they exceeded the decoder size. Even less for the evaluation tasks. We state that now in Section 4.
>
> (6) “Why does making the task harder result in better performance?” We believe that making the pre-training task harder forces the models to develop a deeper understanding (e.g. consider the ELECTRA paper). Also in our setting this intuition seems to apply: Compare “Skip-tree (no <MASK>)” to “Skip-tree (weighted)”.
>
> > (7) Do the authors mean that for each math statement they generate n=100 samples with the <PREDICT> and <MASK> tokens?
>
> Exactly. We now improved the description of the data generation.
>
> > If so, this implies that there will be 360k*100 examples in the larger training set. But this is not consistent with the data stats provided in Table 1.
>
> Thanks for catching this! We failed to describe a filtering step we had applied during data generation. We filter out expressions that are very long >10k characters, because they exceed the size of the encoder so much that the model would only see a small prefix of such an expression. While this filtered out a significant number of training examples (~31%), it only filtered out an insignificant number of evaluation examples (less than 0.16%). We added a description of the filter and we also added an ablation study showing that without the filter the performance on the downstream is worse.
>
> (8) For the missing assumptions and equality completion tasks, we had already provided human readable versions of the ground-truth, but somehow we have forgotten to add them for the type inference and hard type inference tasks. This is now fixed.
>
> (9) Exactly, model updates.

---

### Official Review · AnonReviewer4 · 2020-10-30
**Interesting work**

**Rating:** 7
**Confidence:** 4

**Review:**

This paper extends the idea of language-model style self-supervised learning approach to training logical reasoning models from unlabeled mathematical expressions. The main idea is to develop a skip-tree proxy task (self-supervision) for training the encoder-decoder architecture.  The skip-tree method masks out a complete sub-tree in the input and linearizes it into a sequence in the form of S-expression. The model is required to predict the masked subtree at the decoder end. The paper also proposes several new reasoning tasks for evaluating the model performance. Experimental results show that models learned from this task significantly outperform those trained on the skip-sequence task. Furthermore, the model also exhibits good conjecturing ability in generating quite reasonable amount of new theorems that are provable and useful, which is quite encouraging and impressive.

I’m curious about why the method only masks out one sub-tree for prediction while treating other masked sub-trees as auxiliary part (by increasing the difficulty of the task). This seems to be a waste of self-supervision signals. It might be more (sample) efficient to predict all the masked part in one sample just as what BERT did. It would be helpful to provide experimental justification for this specific design choice if deemed so.

The current pretrained model is used on the newly created tasks without any finetuning because these tasks are similar to the mask prediction problem. This is interesting, but I’m wondering if the proposed method be used in various other downstream reasoning tasks? For example, even with certain finetuning, could the pretrained reasoning model be used in other downstream tasks that are less similar to the pretraining objective? Since it is claimed earlier that ``In contrast, we train language models on an unsupervised proxy task that does not require labeled data and can thus be applied to almost any source of mathematical expressions” (Section 2), it would be necessary to further evaluate the pretrained models on other popular logic reasoning tasks. This would show how generalizable the skip-tree pretrained model is on various other downstream reasoning tasks, which would greatly enhance the strength of the work.

---

> ### Author Response · Authors · 2020-11-17
> **Author response**
>
> Thank you for the review and the great suggestions.
>
> > I’m curious about why the method only masks out one sub-tree for prediction [...]
>
> BERT does not directly apply to encoder-decoder models, so the methodology has to be adapted. In principle, we could concatenate multiple masked-out subexpressions (similar to T5) until we reach the context length of the decoder. This may indeed be an improvement to our current training method.
>
> > I’m wondering if the proposed method can be used in various other downstream reasoning tasks? For example,
> > even with certain finetuning, could the pretrained reasoning model be used in other downstream tasks that are
> > less similar to the pretraining objective?
>
> In this paper we wanted to focus on the finding that self-supervised training alone can lead to interpretable mathematical reasoning abilities, which, we believe, is an astonishing development by itself.
>
> It is very plausible that using skip-tree training as a pre-training and then fine-tune on other reasoning tasks improves the performance on those tasks. However, other papers on the same dataset do not use sequence-to-sequence models, which makes it non-trivial to compare them to our approach. Comparisons on datasets from entirely different sources come with technical problems, such as different tokenization. While it is probably possible to overcome these problems, those discussions would distract from the core insight presented in this paper.

---

### Author Response · Authors · 2020-11-25
**Updated version of the paper**

As requested, we reran the experiments with an improved evaluation methodology. We now report the performance on the test sets instead of the validation sets. We have also changed the learning rate in these experiments (from 5e-5 to 1e-4), as we learned in the meantime that this leads to more stable results. This also led to a notable improvement of the skip-sequence (short) experiment, but none of the conclusions of the paper are affected.

The other requested experiment, to scale up the Transformers, is still in progress, but we have preliminary results. For a model with 110M parameters (in contrast to the 39M used in the other experiments) the performance on equality completion and missing assumption tasks rises to 55% and 51%, respectively, which we find very encouraging.

We also addressed the minor remarks of the reviewers. We will scan the final reviews for updated suggestions.

---

### Decision · Program_Chairs · 2021-01-07
**Final Decision**

**Decision:**

Accept (Spotlight)

**Comment:**

All reviewers find the idea of self-supervised learning on mathematical reasoning with the proposed skip-tree training interesting and gave the firmly positive scores.  The paper is clearly written, and the experiments and the analysis are well-organized, particularly the ability of free-form conjecturing is quite thought-provoking.  Also, the reviewers' initial concerns have been properly addressed during the discussion phrase.

I think this is a good paper from which people can learn a lot, and should be broadly presented at the conference either as an oral or a spotlight presentation.